# Effect of Treatment with Peptide Extract from Beef Myofibrillar Protein on Oxidative Stress in the Brains of Spontaneously Hypertensive Rats

**DOI:** 10.3390/foods8100455

**Published:** 2019-10-06

**Authors:** Seung Yun Lee, Sun Jin Hur

**Affiliations:** Department of Animal Science and Technology, Chung-Ang University, 4726 Seodong-daero, Daedeok-myeon, Anseong-si, Gyeonggi 17546, Korea; seungyun.lee57@gmail.com

**Keywords:** oxidative stress effect, beef peptide extract, brain, spontaneously hypertensive rats

## Abstract

This study was conducted to determine the effect of beef peptide extract on oxidative stress in the brains of spontaneously hypertensive rats (SHRs). A 3-kDa peptide extract was obtained from beef myofibrillar protein using alkaline-AK (AK3K). Oxidative stress in SHR brains was measured by assessing malondialdehyde (MDA) and reactive oxygen species (ROS) concentrations and superoxide dismutase (SOD), catalase, and glutathione peroxidase (GPx) activity. The SHR brains treated with the AK3K peptide extract (400 mg/kg body weight, AK3K400) showed a significant decrease in MDA and ROS contents by 0.33 and 23.92 μM, respectively (*p* < 0.05) compared to the control. The SOD activity for AK3K400 was 61.26%, around 20% higher than the control. Furthermore, the SHRs treated with the AK3K peptide extract showed results similar to those obtained using captopril, a hypertension drug, except for the MDA level. The study demonstrates that the beef peptide extract inhibits the generation of oxidative stress in the SHR brain and could possibly be used for neuronal hypertension therapy.

## 1. Introduction

Hypertension is associated with increased oxidative stress in different organs such as the vascular tissue, brain, and kidney [1,2]. Angiotensin converting enzyme (ACE) inhibitors, including captopril, enalapril, and lisinopril, are mainly used in the treatment of cardiovascular diseases [3]. ACE inhibitors reduce the level of oxidative stress (H_2_O_2_, malondialdehyde (MDA), and reactive oxygen species (ROS) formation) and increase anti-oxidative activity in rats with hyperglycemia, diabetes, hepatic fibrosis, and hypertension [4,5,6]. 

Oxidative stress leading to the production of ROS is considered one of the risk factors for the development of cardiovascular disease, including hypertension, heart failure, and stroke [7,8,9]. Because oxidative stress is closely related to hypertension, decreasing oxidative damage may lead to a reduction in blood pressure and vasodilation [10]. Oxidative stress factors such as lipid peroxide, ROS, and superoxide dismutase (SOD) have been investigated in hypertensive patients and animal studies [11,12]. Experimental studies in rodent models and humans have revealed that the exogenous administration of antioxidants attenuates the hypertensive effect of ROS, resulting in an improvement in vascular function and a reduction of blood pressure [13,14]. The current review suggests that brain health is an important factor in late stage hypertensive disease because of the correlation of hypertension with the aging of the brain [15,16]. A previous study showed long-lasting activity of ACE inhibitors as hypertension agents in spontaneous hypertensive rat (SHR) brains [17].

There has been a growing interest in foods for healthy living, and there have been several reports about bioactive peptides as functional materials. Bioactive peptides derived from food proteins have been studied to show different beneficial effects including those of the antioxidant, ACE inhibitory, neuroprotective, anti-microbial, anti-inflammatory, and anti-tumor types [18,19,20,21]. Specifically, the administration of peptides derived from food sources exerts anti-hypertensive effects on SHRs [22,23,24]. Our previous studies also showed that peptides obtained from beef myofibrillar protein have ACE inhibitory activity and anti-hypertensive effects on SHRs and neuroprotective effects on human neuronal cells [25,26,27]. However, the relationship between oxidative stress in the SHR brain and ACE inhibitory peptide from beef myofibrillar protein has not been reported extensively. Therefore, the purpose of this study was to investigate the effect of peptide extract from beef myofibrillar protein on oxidative stress in SHR brains. 

## 2. Materials and Methods

### 2.1. Materials

All chemicals and reagents were of analytical grade. The lipid oxidation, ROS, reactive nitrogen species (RNS), SOD, and catalase activity kits were purchased from Cell Biolabs (San Diego, CA, USA). The glutathione peroxidase (GPx) assay kit was purchased from Abcam (Cambridge, UK). Phosphate buffered saline (PBS), heparin, ethylenediaminetetraacetic acid (EDTA), tris, butylated hydroxytoluene (BHT) and sodium chloride were purchased from Sigma-Aldrich (St. Louis, MO, USA). 

### 2.2. Preparation of Peptide Extract

The peptide extract was obtained according to the methods of a previous study (Lee and Hur, 2017). Briefly, the hydrolysate obtained from beef myofibrillar protein was processed using alkaline-AK, which activated an enzyme reaction under pH 11 at 60 °C for 8 h. The hydrolysate was then heated under 60 °C for 8 h to stop the enzyme reaction. Then, the hydrolysate was subjected to ultrafiltration using Amicon^®^ Ultra with a molecular mass cut-off of <3 kDa (Millipore, Billerica, MA, USA). The peptide extract with <3 kDa using alkaline-AK (AK3K) was lyophilized at −80 °C for 72 h and stored at −20 °C. 

### 2.3. Animals

The SHR model is considered to be the standard animal model for studying oxidative stress in the brain [28,29]. The SHR model was established by selective inbreeding of Wistar Kyoto rats (WKY). A total of 60 male SHR/Izm (Izumo) (10 weeks old, 20 mice/per experiment) rats were purchased from Central Lab, Animal Inc. (Seoul, Korea). All animals were housed four per cage with food and water available ad libitum and maintained on a 12 h light/dark cycle. All animal experiments were performed following the animal care ethics guidelines with protocols approved by the Animal Care Committee of the KPC Research Co., Ltd. of Korea (P182019). The in vivo experiments were conducted in triplicate. After 1 week of adaptation, the SHRs were randomly divided into four groups, with five rats in each treatment group: T1) Control (distilled water); T2) captopril as the hypertension drug (20 mg/kg body weight); T3) <3 kDa peptide extract obtained by alkaline-AK at 400 mg/kg body weight in distilled water (AK3K400); and T4) <3 kDa peptide extract obtained by alkaline-AK at 800 mg/kg body weight in distilled water (AK3K800). All animals were orally administered drugs in a 1 mL solution using a disposable plastic syringe.

### 2.4. Determination of Antioxidant Enzyme Activities

#### 2.4.1. Analysis of Lipid Oxidation Using the Thiobarbituric Acid-Reactive Substance (TBARS) Assay

The brain samples were homogenized in PBS (pH 7.4) including heparin and 1X BHT for the determination of MDA levels. The MDA in the brains was measured using the Oxiselect thiobarbituric acid-reactive substance (TBARS) assay kit containing thiobarbituric acid-reactive substances (Cell Biolabs, San Diego, CA, USA). Each sample was expressed as micromoles per mg protein according to the manufacturer’s instructions. The total protein concentration was measured before the MDA assay using the Pierce^®^ BCA Protein Assay kit (Thermo, Rockford, USA).

#### 2.4.2. Analysis of ROS Generation

The brain samples (50 mg/mL) were homogenized in PBS (pH 7.4). Then, the samples were centrifuged at 10,000 x *g* for 5 min, and the tissue lysate supernatant was collected. An Oxiselect In Vitro ROS/RNS Assay kit (Cell Biolabs, San Diego, CA, USA) was used to measure the ROS concentration in the brains of the rats according to the manufacturer’s instructions. The ROS or RNS in samples were measured fluorometrically (2′,7′-Dichlorodihydrofluorescein converted to 2′,7′ dichlorofluorescein) against a hydrogen peroxide. 

#### 2.4.3. Analysis of SOD Activity

The brain samples were homogenized in a 1X lysis buffer (10 mM Tris, pH 7.5, 150 mM NaCl, 0.1 mM EDTA). Then, the samples were centrifuged at 1000 x *g* at 4 °C for 10 min, and the tissue lysate supernatant was collected. The SOD activity in the brain was determined using the OxiSelect SOD Assay kit (Cell Biolabs, San Diego, CA, USA).

#### 2.4.4. Analysis of Catalase Activity

The brain samples were homogenized in PBS (pH 7.4) including 1 mM EDTA. Then, the samples were centrifuged at 10,000 x *g* at 4 °C for 15 min, and the tissue lysate supernatant was collected. The catalase activity in the brain was measured using the OxiSelect Catalase Assay kit (Cell Biolabs, San Diego, CA, USA).

#### 2.4.5. Analysis of GPx Activity

The brain samples were homogenized in a cold assay buffer. Then, the samples were centrifuged at 1000 x *g* at 4 °C for 15 min, and the tissue lysate supernatant was collected. The GPx activity in the brain was determined using the GPx Assay Kit (Cell Biolabs, San Diego, CA, USA).

### 2.5. Statistical Analyses

All data are presented as the mean ± standard deviation (SD). All statistical analyses were performed using a one-way analysis of variance using SPSS 20.0 (IBM, Armonk, NY, USA). Tukey’s multiple comparisons test was used to determine the significance of differences between means of different experimental groups, and a *p* value of less than 0.05 was considered statistically significant.

## 3. Results and Discussion

This study was conducted to determine the anti-oxidative stress effect of the peptide extract from beef myofibrillar protein on the SHR brain by examining different oxidative stress factors such as MDA, ROS, SOD, catalase, and GPx activity. 

MDA quantification is frequently used as an indicator of lipid peroxidation by oxidative stress using the TBARS method [30]. The concentrations of MDA were measured as an indication of lipid oxidation in the SHR brain. As shown in Figure 1, the MDA concentration in SHR brains treated with the AK3K peptide extracted at 400 and 800 mg/mL was 0.33 ± 0.09 and 0.40 ± 0.10 μM, respectively, which was significantly lower than that of the control and captopril treatment groups (*p* < 0.05). 

Hypertension can be assessed in the brain because the pathogenesis of hypertension is affected by the activation of the sympathetic nervous system and oxidative stress in the brain [31,32]. Angiotensin II(AngII)increases blood pressure by binding to the Ang II receptor type 1 (AT_1_ receptor), which induces vasoconstriction [33]. In addition, Ang II enhances intracellular oxidative stress, such as increased lipid peroxide, by activating macrophage-mediated oxidation via the AT_1_ receptor [34]. The markers of oxidative stress such as nitrotyrosine, nuclear factor-κB p65 (NF-κB p65), neutrophil cytosol factor 1 (p47phox), and 8-hydroxy-29-deoxyguanosine are significantly elevated in SHRs compared with WKYs [35,36]. In addition, TBARS is increased in the brains of SHRs [37]. Kishi et al. (2004) showed that TBARS levels were significantly higher in different brain tissues, such as the whole brain, rostral ventrolateral medulla, and the nucleus of the solitary tract, compared with the levels in the tissues of WKYs [38]. 

Previous studies have found that ACE inhibitors, including captopril, quinapril, and enalapril, decreased lipid peroxidation and Ang II activation via the thiol group of the ACE inhibitor structure that had free radical scavenging activity [34,39]. Our previous study found that the hydrolysate from myofibrillar protein could chelate the iron in the ACE active center via the carboxylate anion at the C-terminal peptide, resulting in ACE inhibitory and free radical scavenging activity [27]. However, captopril failed to elevate antioxidant properties or inhibit lipid peroxidation in the presence of myoglobin/H_2_O_2_ or iron chloride/ascorbate [40,41]. The reason was speculated to be that it may have stimulated peroxidation in the presence of iron chloride by reducing Fe^3+^ to the Fe^2+^ [40], and the thiol group can produce lipid peroxidation in the presence of Fe^3+^ [42]. Moreover, our previous study also suggested that the MDA level in serum was higher in SHRs treated with AK3K400 than captopril; additionally, the overall results were similar. Therefore, although the mechanism is not exact, the decrease in lipid oxidation in the AK3K peptide extract-treated SHR brain can be explained by the inhibition of Ang II as well as unaffectedness for the presence of iron environment resulting from the ACE inhibitory activity and free radical scavenging activity of AK3K. 

The ROS content was measured to confirm the anti-oxidative stress effect of the AK3K peptide extract on the SHR brain. As shown in Figure 2, the ROS content of SHR brains treated with captopril and AK3K peptide extracts at 400, 800 mg/mL was 23.92 ± 4.56, 23.12 ± 1.00, and 21.71 ± 4.16 μM, respectively. The decrease in ROS content in the SHR brain was significantly greater in the AK3K peptide extract treatment groups compared to that in the control group (*p* < 0.05).

The ROS level is as an indicator of oxidative stress, and it is implicated in cardiovascular disease, including hypertension; in addition, ROS can increase the blood pressure in the heart, brain, vessels, and kidneys [43]. Previous studies focused on Ang II mechanisms revealed that ROS are critical intracellular signaling molecules in cardiovascular diseases such as hypertension [44,45]. Zimmerman and Davisson (2004) also discussed that ROS in the central neural system is implicated in neuro-dysfunction, and the mediation of hypertension depends on Ang II activation [46]. Moreover, neurogenic hypertension depends on Ang II enhancing the messenger RNA (mRNA) and protein of nicotinamide adenine dinucleotide phosphate (NADPH) oxidase subunits, which leads to ROS production in the rostral ventrolateral medulla [47,48]. Apocynin, which is known as an NADPH oxidase inhibitor, aids in preventing the generation of superoxide in humans and protection against oxidative stress in the rostral ventrolateral medulla, resulting in a decrease in blood pressure in SHRs or neurogenic hypertension mediated by Ang II [49]. Tanriverdi et al. (2017) presumed that the possible reason for the effect of apocynin on neuronal hypertension by oxidative stress can be explained by its antioxidant activity [50]. In addition, antioxidants play an important role in reducing ROS and inhibiting NADPH oxidase, and they can be used as hypertension agents [48,51]. Previous studies have reported that peptides and hydrophobic amino acids decrease ROS generation via antioxidant activities such as free radical scavenging and the inhibition of neuronal nitric oxide synthase (NOS) [52]. In our previous studies, peptide from myofibrillar protein was shown to inhibit the ROS level in H_2_O_2_-damaged neuronal cells, which was likely the result of antioxidant activities of the hydrophobic or acidic amino acids. Fisher et al. (2018) reported that the phospholipase A_2_ inhibitory peptide, which was comprised of hydrophobic amino acid residues, inhibited phospholipase A_2_ activity, which is responsible for the activation of NADPH oxidase type 2 (Nox2) via strong binding to cell membrane phospholipids in Ang II-treated lungs [53]. Therefore, in this study, the decrease in ROS content in the SHR brain might be closely related to blocking the NADPH oxidase-derived ROS production through the inhibition of Ang II activation by antioxidant activities and the permeability of hydrophobic amino acids in AK3K peptide extracts. 

As shown in Figure 3, the SOD activity in SHR brains was significantly increased by treatment with AK3K400 and captopril compared to the control (*p* < 0.05). However, the SOD activity of the SHR brain treated with AK3K800 was not significantly different from that in the control group. Previous studies reported that treatment with an excessive concentration could cause toxicity in cell lines through numerous pathways such as the over activation of mechanism receptors or genotoxicity and mutagenicity [54,55]. The oxidative stress induced by toxicity, whether persisting or very high, may cause protein degradation and protein oxidation such as peptide bond cleavage and amino acid oxidation; furthermore, initial antioxidant enzymes also may be damaged [56,57]. Even though the reason for for the lower SOD activity in AK3K800 than AK3K400 has still not been found, we considered that toxicity or oxidative stress by the high concentration treatment may affect the decrease in SOD activity or biological change by the structure change of the AK3K peptide extract. 

The catalase activity in the SHR brain was also determined in this study (Figure 4); it did not show any significant differences between treatment groups. As shown in Figure 5, the GPx activity was also not significantly different between groups. 

SOD, catalase, and GPx are known as representative antioxidant enzymes which are able to regulate oxidative stress and free radicals. SOD plays an important role in catalyzing the decomposition of the highly reactive superoxide anion (-O_2_^−^) to less reactive hydrogen peroxide and oxygen [58]. Catalase is also known to catalyze the conversion of hydrogen peroxide into water and oxygen. GPx, which converts alcohols or free hydrogen peroxide into water, is important for the protection of organisms from oxidative stress [59]. The results of a previous study indicated a relationship between hypertension and antioxidant enzymes [60]. Shou et al. (1997) found that the SOD, GPx, and catalase activity levels in SHRs were lower than those in normal rats, and an ACE inhibitor decreased oxidative stress and increased SOD, GPx, and catalase activity levels in the tissues [61]. Recently, Beltrán-Barrientos et al. (2018) indicated [62] that captopril and food materials did not cause any significant differences in anti-hypertensive effects but did result in a significantly different oxidative stress level compared with the control group. Alternately, in the present study, the catalase and GPx activities of the SHR brains treated with the AK3K peptide extract and captopril were similar to those in the control group. The antioxidant peptide gp91ds, which selectively inhibits group of NAD(P)H oxidase, such as cysteine-arginine-proline-proline-arginine (CRPPR) and cysteine-serine-histidine-methionine-alanine-arginine-threonine-lysine-cysteine (CSHMARTKC) reduced superoxide production stemming from the inhibition of incorporation of NADPH oxidase, resulting in anti-hypertensive activity in stroke-prone SHRs [63]. Similarly, captopril, a sulfhydryl-containing ACE inhibitor, may potentiate antioxidant enzymes by free radical scavenging activity [64]. Carnosine, comprising the amino acids beta-alanine and histidine, has been proven to be a therapeutic candidate for the treatment of degenerative diseases because it can scavenge superoxide and increase superoxide dismutase activity by the metal-ion chelating action [65]. The peptides derived from food protein are composed of hydrophobic amino acids at the N-terminus, which may affect their ability to quench the superoxide radical (O_2_^−^) and hydrogen (H_2_O_2_) [66,67]. We previously reported that the peptides with higher hydrophobic amino acid (leucine and methionine) content showed higher SOD activity and protective effects against H_2_O_2_-treated neuronal cells [25]. In addition, the peptide used in this study had effective anti-hypertensive activity in SHRs and could be used as a therapeutic treatment for cardiovascular disease [26]. Previous studies have reported that the AK3K peptide fraction contains hydrophobic amino acids (valine, leucine, isoleucine), and further AK3K peptide sequences such as threonine–glutamine–lysine–lysine–valine–isoleucine–phenylalanine–cysteine and leucine-isoleucine–valine–glycine-isoleucine–isoleucine–arginine–cystein–valine were mainly identified. This study indicates that the content of other enzymes such as GPx and catalase may not influence the inhibition of oxidative stress in the SHR brain and the treatment of neuronal hypertension. On the other hand, SOD activity in the SHR brain treated with the AK3K peptide extract may have an influence on the regulation of vasoconstriction and the oxidative state, which are mediated by neuronal hypertension. The change in SOD activity in the SHR brain might be due to the interaction of the residue charges of the hydrophobic or acidic amino acids in the AK3K peptide extract, which have a similar mechanism to SOD, such as the reduction and oxidation of the active site of SOD, resulting in a synergy effect by converting the superoxide anion to hydrogen peroxide. Furthermore, the AK3K peptide extract might not further react in the enzymatic defense process because of the lack of interaction between amino acids and these enzymes (GPx and catalase). However, the detailed role of antioxidant enzymes in inhibiting oxidative stress-induced hypertension is still unclear, and it may be affected by amino acid composition or sequence. The results of this study were in agreement with our previous finding that the AK3K peptide (1.25 mg/mL) significantly increased SOD activity, whereas the catalase level was not significantly affected [25]. Ighodaro and Akinloye (2018) also referred to SOD as an important major antioxidant enzyme for protection against neurological disease and hypertension in both humans and animals [68]. 

Therefore, SOD plays a key role in controlling oxidative stress in the SHR brain via the inhibition of ROS and MDA production. Furthermore, this study showed that the peptide extract from beef myofibrillar protein could help to reduce neuronal hypertension through the inhibition of oxidative stress in the brain. 

## 4. Conclusions 

This study showed that the peptide extract from beef myofibrillar protein obtained using alkaline-AK has an anti-oxidative stress effect related to hypertension in the SHR brain. The SHRs treated with AK3K peptide extracts showed a significant decrease in the MDA level and ROS generation compared to the control. The SOD activities in SHRs treated with AK3K peptide extract (400 mg/mL) and captopril were significantly higher compared to those in the other treatment groups. However, the catalase and GPx activities in SHRs treated with each sample were not significantly different between groups. Based on the results of this study, we assume that the anti-oxidative stress property of the AK3K peptide extract can alleviate neuronal hypertension. Furthermore, the AK3K peptide extract could possibly be used in neuronal hypertension therapy. 

## Figures and Tables

**Figure 1 foods-08-00455-f001:**
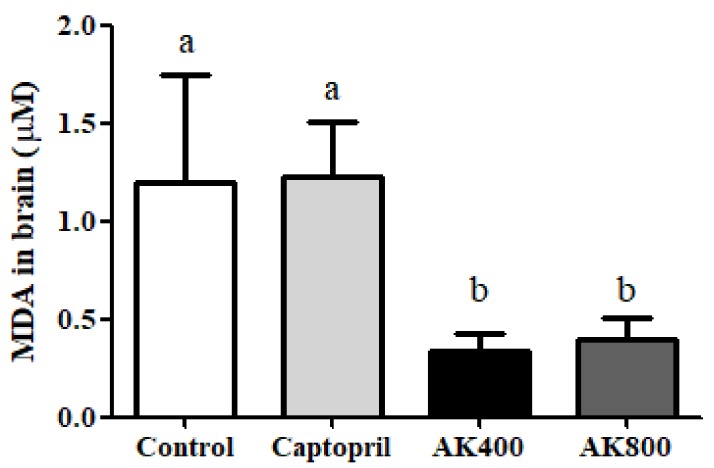
Thiobarbituric acid-reactive substance (TBARS) concentration (µmoL malondialdehyde; MDA) in the spontaneously hypertensive rat (SHR) brain. The groups were the control (distilled water), captopril (20 mg/kg body weight), AK3K400 (400 mg/kg body weight), and AK3K800 (800 mg/kg body weight). Data are presented as the mean ± standard deviation (SD). ^a-b^ Different letters indicate statistically significant differences (*p* < 0.05) in the different groups.

**Figure 2 foods-08-00455-f002:**
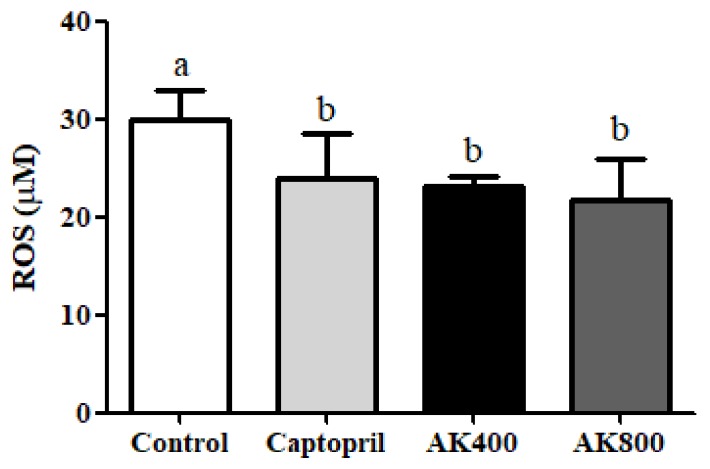
Reactive oxygen species (ROS) generation in the SHR brains. The groups were the control (distilled water), captopril (20 mg/kg body weight), AK3K400 (400 mg/kg body weight), and AK3K800 (800 mg/kg body weight). Data are presented as the mean ± SD. ^a-b^ Different letters indicate statistically significant differences (*p* < 0.05) in the different groups.

**Figure 3 foods-08-00455-f003:**
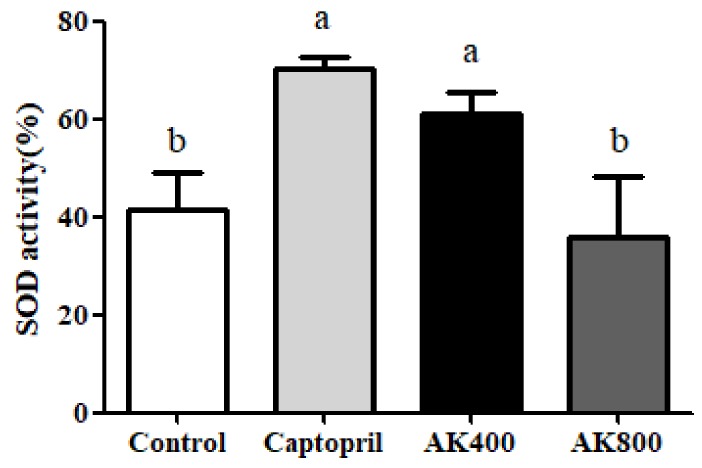
Superoxide dismutase (SOD) activity in the SHR brains. The groups are the control (distilled water), captopril (20 mg/kg body weight), AK3K400 (400 mg/kg body weight), and AK3K800 (800 mg/kg body weight). Data are presented as mean ± SD. ^a-b^ Different letters indicate statistically significant differences (*p* < 0.05) in the different groups.

**Figure 4 foods-08-00455-f004:**
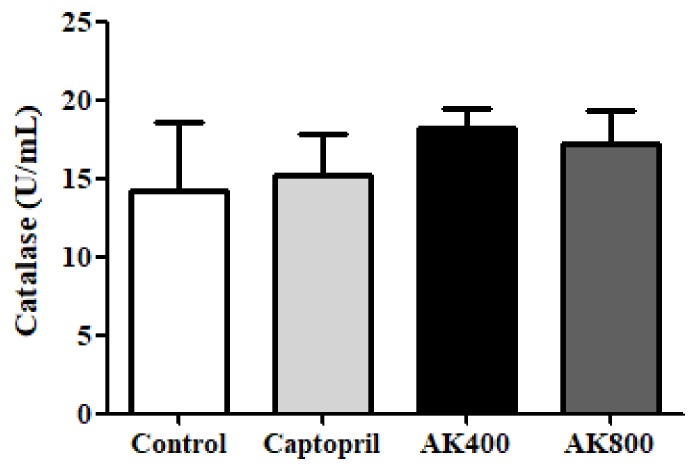
Catalase activity in the SHR brains. The groups are the control (distilled water), captopril (20 mg/kg body weight), AK3K400 (400 mg/kg body weight), and AK3K800 (800 mg/kg body weight). Data are presented as mean ± SD.

**Figure 5 foods-08-00455-f005:**
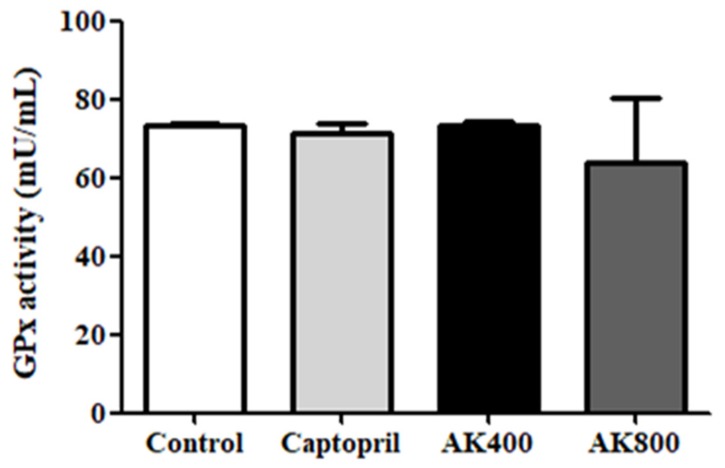
Glutathione peroxidase (GPx) activity in the SHR brains. The groups are the control (distilled water), captopril (20 mg/kg body weight), AK3K400 (400 mg/kg body weight), and AK3K800 (800 mg/kg body weight). Data are presented as mean ± SD.

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
