# Peer review of "Effect of Treatment with Peptide Extract from Beef Myofibrillar Protein on Oxidative Stress in the Brains of Spontaneously Hypertensive Rats"

_foods, 2019, doi:10.3390/foods8100455_

Round 1

Reviewer 1 Report

Manuscript ID: Foods-557868

The manuscript study the effect of a peptide extract from beef myofibrillar proteins obtained from alkaline-AK hydrolysis on the oxidative stress in brains of SHRs. The manuscript is well written but simple, because only antioxidant activities were measured using enzymatic kits.

Some considerations are listed below:

-          The study was done using a peptide extract obtained from myofibrillar proteins by alkaline-AK treatment but the methodology for sample preparation is not described. Please, explain or reference in “materials and methods”.

-          I think the word “peptide” used throughout the manuscript is not correct, as the sample is not a single peptide but a peptide extract or peptide hydrolysate from beef myofibrillar proteins.

-          Line 16: please rewrite regarding the SOD activity because is confusing. The SOD activity for AK3K was 61.26%, around 20% higher than control.

-          Line 17-18: results were not similar in MDA levels.

-          Lines 129-136: could you explain better a possible explanation to get different results for captopril and AK3K samples?

-          Line 175: could you give a reason why the SOD activity of AK3K800 was different to AK3K400 and similar to control?

-          Line 194: Define CAT previously and be consistent with this abbreviation, because it has mainly been used “catalase” throughout the text.

-          Line 231: AK3K peptide extract (400mg/mL)

-          Lines 242-243: “a similar result was also shown when the SHRs were treated with the ACE inhibitor captopril” is confusing. Significant differences were found in the MDA level between AK samples and captopril, whereas no significant differences were found for ROS generation.

-          Line 244 is also confusing, please rewrite.

-          Lines 245-248: Both sentences seem to say the same.

-          Figures 4 and 5: Letters indicating statistically significant differences were lost.

-          It would be interesting the identification of the peptides present in the extracts in order to study those potentially responsible for the observed antioxidant activities.

Author Response

<Reviewer 1>

Q1:The study was done using a peptide extract obtained from myofibrillar proteins by alkaline-AK treatment but the methodology for sample preparation is not described. Please, explain or reference in “materials and methods”.

A1: Thank you for your review and we agree with your comment. We described prepration of peptide extrac in “materials and methods”.

→ p. 2 line 62-69, “2.2 Preparation of peptide extract” 

Q2: I think the word “peptide” used throughout the manuscript is not correct, as the sample is not a single peptide but a peptide extract or peptide hydrolysate from beef myofibrillar proteins.

A2: Thank you for your review and we agree with your comment. We revised “peptide” to “peptide extract” in manuscript. 

Q3:Line 16: please rewrite regarding the SOD activity because is confusing. The SOD activity for AK3K was 61.26%, around 20% higher than control.

A3: Thank you for your review and we agree with your comment. We rewrote the sentence to avoid confusion.

→ p. 1 line 16-18, “MDA and ROS contents by … 20% higher than control.”

Q4: Line 17-18: results were not similar in MDA levels.

A4: Thank you for your review and we agree with your comment.

We revised the sentence according to your comment.

→ p. 1 line 19, “…except for MDA level.” 

Q5: Lines 129-136: could you explain better a possible explanation to get different results for captopril and AK3K samples?

A5: Thank you for your review and we agree with your comment. We explained better a possible planation.

→ p. 4 line 143-156, “Previous studies found that ACE inhibitors … free radical scavenging activity of AK3K”

Q6: Line 175: could you give a reason why the SOD activity of AK3K800 was different to AK3K400 and similar to control?

A6: Thank you for your review and we agree with your comment. We described according to your comment.

→ p. 5-6 line 193-203, “As shown in Fig. 3, … the structure change of AK3K peptide extract.”

Q7: Line 194: Define CAT previously and be consistent with this abbreviation, because it has mainly been used “catalase” throughout the text.

A7: Thank you for your review and we agree with your comment. We revised “CAT” to “catalase” in manuscript.

→ p. 7 line 220, 223.

→ p. 8 line 275.

Q8: Line 231: AK3K peptide extract (400mg/mL)

(AK3K peptide extract)

A8: Thank you for your review and we agree with your comment. We revised the sentence.

→ p. 8 line 261-262, “the AK3K peptide (1.25 mg/mL) … significantly affected [25].”

Q9: Lines 242-243: “a similar result was also shown when the SHRs were treated with the ACE inhibitor captopril” is confusing. Significant differences were found in the MDA level between AK samples and captopril, whereas no significant differences were found for ROS generation.

A9: Thank you for your review and we agree with your comment. We revised the sentence to avoid confusion.

→ p. 8 line 271-273, “The SHRs treated with AK3K peptide extract … generation compared to the control.”

Q10: Line 244 is also confusing, please rewrite.

A10: Thank you for your review and we agree with your comment. We revised to avoid confusion.

→ p. 8 line 273-276, “The SOD activity in SHRs treated … samples were not significantly different between groups.”

Q11: Lines 245-248: Both sentences seem to say the same.

A11: Thank you for your review and we agree with your comment. We revised to avoid same sentences.

→ p. 8 line 276-278, “Based on the results of this study, … in neuronal hypertension therapy.”

Q12: Figures 4 and 5: Letters indicating statistically significant differences were lost.

A12: Thank you for your review and we agree with your comment.

We deleted the sentence indicating significant differences in figure 4 and 5.

→ Figure 4, line 215.

→ Figure 5, line 219.

Q13: It would be interesting the identification of the peptides present in the extracts in order to study those potentially responsible for the observed antioxidant activities. 

A13: Thank you for your review and we agree with your comment.

We wrote the information about the identification of the peptide extract in the manuscript based on our previous study .

→ p. 7, line 246-248.”The AK3K peptide fraction contained… Leu-Ile-Val-Gly-Ile-Ile-Arg-Cys-Val [25,26]. ”

<Reviewer 2>

Q1: Information about the peptide AK3K is not mentioned in this manuscript. It is important data for better support of the discussion in page 7. 

A1: Thank you for your review and we agree with your comment. We wrote the information about the AK3K peptide extract in manuscript.

→ p.2 line 62-69, 2.2 Preparation of peptide extract

→ p.7 line 246-248, “The AK3K peptide fraction contained…Leu-Ile-Val-Gly-Ile-Ile-Arg-Cys-Val [25,26].”

Q2: ROS is a multi-component, so which component is being measured in the assay used. Principle of the assay should be described to know what is being measured (peroxide, superoxide, hydroxyl radicals…etc)

A2: Thank you for your review and we agree with your comment. We described more information about principiple of the assay.  

→ p.3 line 96-98, “The ROS or RNS in samples … against a hydrogen peroxide.”

Q3: Introduction-line 33; Vasoconstriction or Vasodilation? It seems you mean Vasodilation

A3: Thank you for your review and we agree with your comment. We revised the wrong word.

→ p.1 line 34, “…vasodilation [10].”

Q4: Method-line 74; What is BHT?

A4: Thank you for your review. We wrote about BHT.

→ p.2 line 59, “butylated hydroxytoluene (BHT)”

Q5: Results-line 111; Is the SD (±99) correct?

A5: Thank you for your review. We revised wrong results.

→ p.3 line 125, “0.33 ± 0.09”

Reviewer 2 Report

The manuscript discusses the in vivo antioxidative stress activity in SHR rats of a peptide originating from beef myofibrillar protein. It is a follow up study of the peptide that it had anti-hypertensive effects on SHR. This paper describes an interesting finding that the peptide exert their antihypertensive activity through its antioxidant modulatory action.

As a whole, the manuscript is well written and makes some interesting points that further illuminate the importance of the peptide as nutraceutical agents.  The text focuses well on its target points and balances with the figures included. 

However, I have some concerns:

Information about the peptide AK3K is not mentioned in this manuscript. It is important data for better support of the discussion in page 7. ROS is a multi-component, so which component is being measured in the assay used. Principle of the assay should be described to know what is being measured (peroxide, superoxide, hydroxyl radicals…etc) Introduction-line 33; Vasoconstriction or Vasodilation? It seems you mean Vasodilation Method-line 74; What is BHT? Results-line 111; Is the SD (±99) correct?

Author Response

<Reviewer 2>

Q1: Information about the peptide AK3K is not mentioned in this manuscript. It is important data for better support of the discussion in page 7.

A1: Thank you for your review and we agree with your comment. We wrote the information about the AK3K peptide extract in manuscript.

→ p.2 line 62-69, 2.2 Preparation of peptide extract

→ p.7 line 246-248, “The AK3K peptide fraction contained…Leu-Ile-Val-Gly-Ile-Ile-Arg-Cys-Val [25,26].”

Q2: ROS is a multi-component, so which component is being measured in the assay used. Principle of the assay should be described to know what is being measured (peroxide, superoxide, hydroxyl radicals…etc)

A2: Thank you for your review and we agree with your comment. We described more information about principiple of the assay.

→ p.3 line 96-98, “The ROS or RNS in samples … against a hydrogen peroxide.”

Q3: Introduction-line 33; Vasoconstriction or Vasodilation? It seems you mean Vasodilation

A3: Thank you for your review and we agree with your comment. We revised the wrong word.

→ p.1 line 34, “…vasodilation [10].”

Q4: Method-line 74; What is BHT?

A4: Thank you for your review. We wrote about BHT.

→ p.2 line 59, “butylated hydroxytoluene (BHT)”

Q5: Results-line 111; Is the SD (±99) correct?

A5: Thank you for your review. We revised wrong results.

→ p.3 line 125, “0.33 ± 0.09”

Round 2

Reviewer 1 Report

The manuscript has been improved considerably according to the reviewers comments.

However, some changes should still be considered:

line 195: "could cause toxicitiy..."

line 197-198: "it is very high, may cause..."

line 200: "is still not fond, ..."

line 201: "treatment may affect ..."

line 245-247: " Previous studies reported that the AK3K peptide fraction
contained hydrophobic amino acids (Val, Leu, Ile), and further AK3K peptide sequences such as Thr-Gln-Lys-Lys-Val-Ile-Phe-Cys and Leu-Ile-Val-Gly-Ile-Ile-Arg-Cys-Val were mainly identified."

line 271: "The SOD activities..."

Author Response

Q1: line 195: "could cause toxicitiy..."

A1: Thank you for your review and we agree with your comment. We revised the sentence according to your comment.

→ p. 5 line 196, “..could cause toxicity in cell…” 

Q2: line 197-198: "it is very high, may cause..."

A2: Thank you for your review and we agree with your comment. We revised the sentence according to your comment.

→ p. 5-6 line 198-199, “it is very high, may cause...”

Q3: line 200: "is still not fond, ..."

A3: Thank you for your review and we agree with your comment. We revised the sentence according to your comment.

→ p. 6 line 201, “is still not found, ...”

Q4: line 201: "treatment may affect ..."

A4: Thank you for your review and we agree with your comment. We revised the sentence according to your comment.

→ p. 6 line 202, “treatment may affect ...”

Q5: line 245-247: " Previous studies reported that the AK3K peptide fraction
contained hydrophobic amino acids (Val, Leu, Ile), and further AK3K peptide sequences such as Thr-Gln-Lys-Lys-Val-Ile-Phe-Cys and Leu-Ile-Val-Gly-Ile-Ile-Arg-Cys-Val were mainly identified."

A5: Thank you for your review and we agree with your comment. We revised the sentence according to your comment.

→ p. 7 line 246-248, “Previous studies reported that the AK3K peptide fraction... were mainly identified.”

Q6: line 271: "The SOD activities..."

A6: Thank you for your review and we agree with your comment. We revised the sentence according to your comment.

→ p. 8 line 273, “The SOD activities…”
